# Work as a Social Determinant of Racial Health Inequalities

**DOI:** 10.3390/ijerph19169820

**Published:** 2022-08-09

**Authors:** Shannon C. Montgomery, Joseph G. Grzywacz

**Affiliations:** Department of Human Development and Family Science, Florida State University, Tallahassee, FL 32306, USA

**Keywords:** health, mental health, organizational psychology, racial inequalities, social determinants, well-being, work

## Abstract

Interdisciplinary research posits that work is a social determinant of health contributing to racial inequalities in death, disease, and well-being amongst Black individuals in the United States. This study aims to advance research by integrating two theoretical frameworks (Warr’s Vitamin Model and Assari’s “differential exposure” and “differential gain” mechanisms) to investigate the role of work in eudemonic well-being. We included a nationally representative sample of adults who participated in the Midlife in the United States (MIDUS) Refresher and Milwaukee Refresher projects in 2011–2014, alongside corresponding occupational information (O*NET 17.0). The results of this study indicated that three of nine studied job characteristics systematically differ by race. We found evidence of differential gain by race on psychological well-being. Job characteristics had either benign or negative associations with well-being among Black individuals but consistently positive associations with well-being among non-Black individuals. In contrast to Warr’s Vitamin Model, we found little evidence of curvilinear health effects of job characteristics (only 5.5% were statistically significant). Finally, it was found that advanced educational attainment benefited multiple dimensions of well-being among Black individuals but had benign or negative implications for non-Black individuals, after controlling for demographics. Overall, the results highlight racial inequalities in eudemonic well-being because Black individuals face challenges in obtaining jobs that are beneficial to well-being. Collectively, the results reinforce the idea that work is a social determinant of health.

## 1. Introduction

Substantial discipline-specific and interdisciplinary research suggests that work is a social determinant of health [1,2] or a socio-structural attribute that reproduces health inequalities [1,2], including racial inequalities in death, disease, and well-being. Labor sociologists and labor economists have studied racial differences in labor force participation [3], and the elevated burden of un/underemployment and job displacement among African Americans [4]. Economists note that African Americans and other racial and ethnic minorities are disproportionately represented in specific macro- and micro-occupational sectors [5,6] and occupational epidemiologists note that racial and ethnic minorities are often over-represented in the most dangerous occupations [7,8] and experience more workplace injuries and fatalities [9]. Reviews of the interdisciplinary literature focused on specific features of jobs associated with health outcomes note African Americans have fewer opportunities than Whites to exert control or decision making over how their job is performed, and often reported greater exposure to general job stressors and racial discrimination on their jobs [10].

Work as a social determinant of health inequalities, including racial inequalities, remains under researched, despite apparent evidence of its potential and proclamations [1]. Indeed, the U.S. National Institutes of Health’s release of a funding opportunity announcement in the summer of 2021 targeting work as a social determinant of health is evidence of a need for the systematic development and testing of systematic thinking. Like a multifaceted jewel, several features of work have potential to enhance or degrade health [11]. Structural features of work have health potential. Involuntary loss of work (i.e., becoming unemployed) predicts physical and mental health declines [12,13], and the strong connection between employment and health insurance [14] in the United States (U.S.) suggests employment conditions allow access to high-quality health care. Temporal features of work have health potential. The World Health Organization classifies rotating shift work as a probable carcinogen [15] and evidence indicates that workers in temporary or seasonal forms of work experience poorer health outcomes [16]. Work provides a sense of personal identity, social value or prestige, and a framework for organizing time—all of which Jahoda [17] argued have mental health implications. Finally, the diverse array of expressions in how “work is organized” has substantial potential to promote and degrade human health [18].

The health and health inequality implications of all facets of work just described are worthy of scientific inquiry, but this paper focuses on selected indicators of work organization, specifically the attributes and skills required of workers by their jobs. The focus on attributes and skills required of workers by jobs is motivated by both conceptual and practical features. There are several conceptual models in the occupational health psychology literature that emphasize job characteristics that typify allowances or requirements of jobs. The Job Demands-Control Model [19] draws attention to the amount of control or decision-making workers are allowed to exercise over their tasks, along with the psychological demands imposed on workers. Likewise, Warr’s Vitamin Model [20] presents a way of thinking about how features of jobs may contribute to health outcomes. He proposed that just as the deficiencies and excess of certain vitamins impact health negatively, so do certain features of jobs [21]. Warr outlined nine features of jobs shared by several models of occupational stress [19,22,23] that affect health, but then classifies each feature of jobs in two distinct ways to characterize their relationship to health. He hypothesized that some job characteristics, such as salary or compensation, have a “constant effect” pattern—essentially a basic linear association wherein increases continue to offer further health benefits. However, Warr hypothesized that some attributes of jobs, such as productivity expectations or external standards, have non-linear associations with health. That is, just as too much of some vitamins can result in toxicity, things such as productivity expectations are inspiring or motivational, but there is a breaking point—or a point of additional decrement—when they become onerous or “stressful.” More recent tests of the model found support for several of the Vitamin Model’s posited curvilinear relationships between job characteristics (i.e., job demands, job autonomy, and workplace social support) with indicators of employee well-being [24,25]. In addition, a more recent iteration of the model applied the Vitamin Model to mental health and happiness in the broader environment, and found support for a linear (i.e., availability of money, physical security) and curvilinear (i.e., personal control, externally-generated goals) relationship between environmental features and mental health [26].

The general “work organization” orientation and a specific focus on abilities and skills required of workers by jobs is also practically motivated. In the U.S. these features of work have been monitored for decades by the U.S. Employment Service through the Occupational Information Network (O*NET). O*NET and its predecessor (i.e., the Dictionary of Occupational Titles) were designed to facilitate the matching of jobs with workers by collecting systematic occupational information about diverse jobs and the attributes of individuals that enable success in those jobs [27]. Importantly, the O*NET data are publicly accessible, offering characterizations of hundreds of job attributes on nearly 1000 distinct occupations contained in the Standard Occupational Classification. Although the O*NET measurement battery is **not** organized by Warr’s Vitamin Model or any other model of health, it assesses several concepts that fit within these models. A recent study using data from the O*NET explored the association between job control (i.e., having freedom to set goals, make decisions, and apply new knowledge to the job) and self-rated health by race [28]. The results found that for White men, job control was health-protective (i.e., higher job control was associated with lower odds of reporting poor health) but for racial/ethnic minority men, higher job control was associated with higher odds of reporting poor health. Similar findings were reported for women. However, the authors suggested that workers of color are overrepresented in low-control jobs in the U.S. work force, calling for the investigation of variation in O*NET characteristics by race.

Assari argues there are two mechanistic processes by which social determinants contribute to health inequalities, including racial inequalities in health [10]. The first process, “differential exposure,” argues that African Americans and other racial and ethnic minorities have elevated “exposure” to various pathogens, be they physical (e.g., COVID-19), social (e.g., overt discrimination), and psychological (e.g., perceived stress). As it is applied to “work” or “jobs,” the differential exposure mechanism would argue that racial and ethnic minorities are more likely to experience job instability (e.g., unemployment or job displacement), an idea that is evident in the research [29,30]. Among those that are employed, the differential exposure mechanism would argue that racial and ethnic minorities will confront greater exposure to physical, social, and psychological pathogens while working, and less access to the health-enhancing features of work. Landsbergis and colleagues’ [31] systematic review of the occupational health literature supports the differential exposure hypothesis. Nevertheless, the results of that review are limited by the almost exclusive use of self-reported assessments of job features or characteristics; the O*NET allows an opportunity to test differential exposure by race with “objective” measures of job attributes.

The second mechanism by which social determinants contribute to health inequalities was labeled “diminished gain” by Assari. “Diminished gain” argues that racial and ethnic minorities gain fewer health benefits than non-Hispanic Whites to various social resources, including education as a developmental tool for marketable human capital and employment as a means of social productivity. As it is applied to the world of “jobs” or “employment,” the diminished gain mechanism would argue that educational attainment is less effective for racial and ethnic minorities than for Whites in the acquisition of stable, safe, or rewarding employment, a hypothesis that has been supported by previous research [32]. Further, the differential gain hypothesis would argue that desirable attributes or features of work, such as the ability to control how or where work is done or the ability to acquire new skills, have less health-promotive potential for racial and ethnic minorities than for Whites. Although “differential exposure” and “diminished gain” are conceptually distinct, they are likely synergistic and provide a framework for identifying targets for eliminating racial inequalities in health [33].

Previous research found that minority status was a positive predictor of eudemonic well-being, suggestive of the overcoming of race-related hardship [34]. Eudemonic wellbeing suggests that negative experiences and emotion may contribute to one’s life purpose and engagement in life [35]. Furthermore, it emphasizes that life difficulties may contribute to deeper meaning in life, which may improve one’s wellbeing through strengthening social ties, improving self-regard for oneself, and heightened mastery [35]. Contrary to hedonic measures of well-being, Ryff’s six dimensions of well-being encompass the breadth of eudemonic well-being, including positive evaluations of oneself and one’s life, a sense of personal growth and development, belief that life is purposeful and meaningful, the possession of good relationships with other people, the capacity to manage one’s life and the surrounding environment, and a sense of self-determination [36].

The goal of this study is to advance understanding about work as a social determinant of racial inequalities in health. We study eudemonic well-being because mental health may be more sensitive to job attributes in the 21st-century economy than physical health, and some suggest it is a valuable tool for overcoming race-related hardship [34]. In this study, we integrate two theoretical frameworks. First, we apply Warr’s Vitamin Model and the presumption that job attributes and skills can have both “continuing effects” and “additional decrement” implications for health. The job attributes of the Vitamin Model are conceptualized as agents that can affect health, and we use Assari’s “differential exposure” and “differential gain” mechanisms to evaluate their potential for understanding racial inequalities in eudemonic well-being. More formally, this study pursues three main objectives:Investigate racial variation in O*NET job characteristics controlling for other aspects of social stratification (age, sex, education) as an evaluation of differential exposure.Investigate the extent to which features of jobs (i.e., O*NET characteristics) have “constant effect” (linear) or “additional decrement” (curvilinear) associations with eudemonic well-being.Investigate whether the associations of educational attainment, as the fundamental determinants of type of employment, and O*NET job characteristics with eudemonic well-being differs by race as an evaluation of differential exposure.

## 2. Materials and Methods

### 2.1. Sample Characteristics

Data for this study were obtained from a nationally representative sample of adults who participated in the Midlife in the United States (MIDUS) Refresher (MR) and Milwaukee Refresher (MKER) projects in 2011–2014. These projects were intended to replenish the original MIDUS 1 project, a national longitudinal study that aimed to investigate the psychological and social factors that may account for age-related variations in health. Participants were recruited into the study via an initial 45-min telephone interview and were invited to complete a mail questionnaire and interview via telephone. Survey data were collected on demographic and physical and mental health information via a 30-min phone interview, followed by two 50-page mailed self-administered questionnaires (SAQ). For MR, the overall response rate for the SAQ was 73% and the phone interview was 71%. For MKER, the overall response rate for the SAQ was 58.9% and the phone interview was 39.8%. From 2012–2013, the MIDUS Milwaukee Refresher study recruited an over-sample of African American adults residing in Milwaukee, WI. Milwaukee respondents were interviewed in their homes using a 2.5-h Computer Assisted Personal Interview (CAPI) protocol and afterwards completed a SAQ. Data on demographics were obtained via self-reporting. Data from the sample were restricted to participants who responded ‘yes’ to the question ‘are you currently working for pay?’. Relevant data from MR was merged with data from MKER, and variables were constructed that contained values from both cohorts. A ‘sample’ variable was created which identified whether participants were from MR or MKER.

### 2.2. Measures

#### Demographic Characteristics

The racial groups were dichotomized into Black individuals (0) or non-Black individuals (1). The number of chronic conditions in the past 12 months was measured by a self-reported continuous scale. Based on the upper quartile range, participants who had four or more chronic conditions were dichotomized as having co-morbid chronic conditions, whereas three or fewer chronic conditions was dichotomized as 0 co-morbid chronic conditions. Control variables included self-reported age (continuous), sex, and educational attainment. The highest level of education completed was categorized into High School/GED or less (0), some college (1), bachelor’s degree (2), and advanced degree (Master’s or PhD) (3).

### 2.3. Psychological Well-Being

Psychological wellbeing is a 42-item composite score composed of autonomy (7-items), environmental mastery (7-items), personal growth (7-items), positive relations with others (7-items), purpose in life (7-items), and self-acceptance (7-items). A six-point scale was used for all items, ranging from 1 (totally disagree) to 6 (totally agree). Psychological well-being scales were constructed by calculating the sum of each set of items. Higher scores reflected higher standings in the scale. For an item with a missing value, the mean value of completed items is imputed. Cronbach’s alpha for autonomy was 0.717 for MR and MKER. Cronbach’s alpha for environmental mastery was 0.804 for MR and MKER. Cronbach’s alpha for personal growth was 0.733 for MR and MKER. Cronbach’s alpha for positive relations with others was 0.789 for MR and MKER. Cronbach’s alpha for purpose in life was 0.741 for MR and MKER. Cronbach’s alpha for self-acceptance was 0.856 for MR and MKER.

### 2.4. O*NET Job Characteristics

This manuscript is the first product of a multi-year project to harvest and append O*NET data to all the survey components of the MIDUS enterprise, including MR and MKER. The O*NET datasets used in this analysis are from 2012, the closest O*NET data release to the periods of data collection for MR and MKER (O*NET 17.0). Over 30 variables were constructed across the O*NET datasets (for a detailed explanation of variable construction and psychometric evaluation see data documentation) [37]. To minimize the number of variables in this analysis, we eliminated constructed variables from the O*NET measurement battery that had strong (r > 0.80) correlations with other constructed variables, resulting in nine variables studied in this analysis. Recall, these assessments are standardized for purposes of job matching, not to capture theoretical concepts from models of work and health. Therefore, for each constructed variable we provide the variable name, a description of the types of items used to construct the variable, and then we align that attribute with one of the job attributes from the Vitamin Model [20] and characterize it as either a constant effect or additional detriment.

*Sensory abilities* is a measure consisting of 12 abilities required by workers to perform their job. The items are from the abilities dataset, and they ask about the importance of primarily visual (e.g., near vision, glare sensitivity, depth perception, peripheral vision), and auditory (e.g., auditory attention, sound localization) abilities to perform the job (α = 0.87). High levels of these attributes tend to center in occupations in the Natural Resources, Construction, and Maintenance occupational sector as well as the Production, Transportation, and Material Moving Sectors and in jobs characterized as routine and non-routine labor [38]. Because sensory abilities are required of workers, particularly workers in manual labor jobs, they are perhaps best characterized as “externally generated goals” which the vitamin model posits as having a curvilinear association with health outcomes.

*Resource management skill* is a measure consisting of four items from the skills dataset. The items capture the importance of managing financial, personal, material, and temporal resources in performing the job (α = 0.88). The O*NET data collection instrument characterizes all the assessed skills, including resource management skills as something that “develops over time through training or experience.” Such a characterization aligns with the “opportunity for skill use and acquisition” domain of the Vitamin Model, which is posited as having a curvilinear association with health outcomes.

*Technical skills* is a measure consisting of eleven items from the skills dataset reflecting the importance of skills related to the design, set up, operation, and correcting of malfunctions in machines or technological systems (α = 0.90). As with other skills, it is presumed to develop over time, thereby aligning it with the “opportunity for skill use and acquisition” domain of the Vitamin Model posited to have a curvilinear association with health outcomes.

*Information input* is a measure consisting of five items from the work activities dataset. The items assess the importance of activities around where and how information and data are collected (e.g., observing or receiving from sources, identifying or detecting changes in information, and monitoring or reviewing information for problems) (α = 0.68). High levels of these attributes tend to center in the Management, Business, Science, and Arts occupational sector and to characterize analytical jobs [38]. The prestige of jobs in this sector and the value placed on analytic abilities aligns with the “valued social position” domain of the Vitamin Model which posits a linear, continuing effect with health outcomes.

*Interacting with others* is a measure consisting of seventeen items from the work activities dataset. The items assess what interactions with other persons or supervisory activities occur while performing the job, (e.g., coaching and developing others, developing and building teams, resolving conflicts, and monitoring and controlling resources) (α = 0.928). These jobs align with the “contact with others” domain of the Vitamin Model which posits a curvilinear relationship with health outcomes.

*Work output* is a measure consisting of nine items from the work activities dataset. The items assess what physical activities are performed, what equipment and vehicles are operated/controlled, and what complex/technical activities are accomplished as job outputs, (e.g., controlling machines and processes, interacting with computers, operating vehicles, and repairing and maintaining electronic equipment) (α = 0.729). These jobs align with the “opportunity for skill use and acquisition” domain of the Vitamin Model which posits a curvilinear relationship with health outcomes.

*Interpersonal relationships* is a measure consisting of fourteen items from the work context dataset. The items describe the context of the job in terms of human interaction processes, (e.g., coordinate or lead others, deal with unpleasant or angry people, public speaking, and work with a work group or team) (α = 0.929). Similar to *interacting with others*, these jobs align with the “contact with others” domain of the Vitamin Model which posits a curvilinear relationship with health outcomes.

*Physical work conditions* is a measure consisting of thirty items from the work context dataset. The items describe the work context as it relates to the interactions between the worker and the physical job environment (e.g., cramped work space, exposed to hazardous equipment, exposed to weather, sounds, and wearing safety equipment) (α = 0.929). These jobs align with the “environmental clarity” of the Vitamin Model which posits a curvilinear relationship with health outcomes.

*Structural job characteristics* is a measure consisting of thirteen items from the work context dataset. The items describe the work context as it involves the relationships or interactions between the worker and the structural characteristics of the job, (e.g., duration of the typical work week, importance of being exact or accurate, level of competition, and time pressure) (α = 0.65). Similar to physical work conditions above, these jobs align with the “environmental clarity” of the Vitamin Model which posits a curvilinear relationship with health outcomes.

### 2.5. Data Analysis

All data analysis was conducted using SPSS Version 27.0 (SPSS Inc., Chicago, IL, USA). Generalized estimating equations accounted for clustering by occupation codes. The first model used generalized estimating equations to investigate racial variation in job characteristics (sensory abilities, information input, work output, interacting with others, technical skills, resource management skills, interpersonal relationships, physical work conditions, and structural job characteristics). An interaction variable was computed between race and education. The second model used generalized estimating equations to predict six dimensions of psychological well-being (autonomy, environmental mastery, personal growth, positive relations, purpose, and self-acceptance) from job characteristics. Job characteristics were mean-centered. Education was dummy-coded into high school or less, some college education, Bachelor’s degree, and Masters or PhD. Interaction variables were computed between race and job characteristics and race and education. The models were tested for linear and non-linear effects. Whilst models were adjusted for age, sex, and sample (MR or MKER), they were not adjusted for income. This is due to the hypothesis that income is a mediator of racial variation in psychological well-being, rather than a confounder of racial variation in psychological well-being, because income follows from, rather than precedes, employment.

## 3. Results

### 3.1. Demographic Characteristics

Demographic characteristics for the respondents included in this study are outlined in Table 1. This study included 2177 individuals (n = 1781 non-Black individuals; n = 396 Black individuals). On average, non-Black individuals were older than Black individuals (45.4 years, 42.2 years, respectively, *p* < 0.017). Non-Black individuals were predominantly male (52.1%) whereas Black individuals were predominantly female (59.1%). The majority of non-Black individuals had a bachelor’s degree or higher (52.7%) whereas Black individuals had on average lower education (38.4% had some college education). On average, Black individuals had higher psychological well-being scores compared to White individuals across all six dimensions (autonomy, environmental mastery, personal growth, positive relations with others, purpose in life, and self-acceptance). However, only autonomy (*p* < 0.001), personal growth (*p* = 0.05) and purpose in life (*p* = 0.006) had statistically significant differences between Black individuals and non-Black individuals. Non-Black individuals had a statistically higher average income ($61,448.04) compared to Black individuals ($39,473.40, *p* < 0.001).

### 3.2. Differential Exposure to Job Attributes

Results from generalized estimating equations that investigated racial variation in job characteristics are outlined in Table 2. Statistically significant racial group differences were observed for three out of nine occupational characteristics, holding education, sex, age, and sample (MR or MKER) constant. Compared to Black individuals, non-Black individuals had greater technical skills (0.117, *p* < 0.05), resource management skills (0.113, *p* < 0.05), and structural job characteristics (0.069, *p* < 0.05). There were no statistically significant differences by race for sensory abilities, information input, interacting with others, interpersonal relationships, work output, and physical work conditions.

Table 3 and Table 4 summarize results obtained from a series of results elaborated in Appendix A (Table A1, Table A2, Table A3, Table A4, Table A5, Table A6, Table A7, Table A8 and Table A9). Table 3 visually depicts all the instances wherein race modified either the linear or curvilinear association of job characteristics with dimensions of well-being. Likewise, Table 4 depicts all the instances where race modified the association of education, a critical indicator of socioeconomic status and human capital brought to the labor market, with well-being. The results showed that psychological well-being increased for non-Black individuals in jobs where sensory abilities, work output, resource management skills, and physical work conditions were more important to job performance, but decreased for Black individuals. However, for jobs were interacting with others, interpersonal relationships and structural job characteristics were more important for job performance, well-being was increased for Black individuals but there was no significant association with Whites’ well-being. Only information input and technical skills were not associated with psychological health for Black individuals nor Whites. Furthermore, only interacting with others and physical work conditions showed some evidence of a curvilinear relationship.

### 3.3. Diminished Gain of Education

Table 4 shows a direct comparison of race as a moderator of the relationships between education and the 6 dimensions of well-being, across the 9 models of job characteristics. Table A1, Table A2, Table A3, Table A4, Table A5, Table A6, Table A7, Table A8 and Table A9 show that across all job characteristics, increasing education increased well-being for Black individuals for all models (with the exception of physical work conditions and self-acceptance), however for Whites, increasing education had a null, lesser, or non-significant effect. Compared to High School Education or GED or below, obtaining certain thresholds of education was a statistically significant predictor of seven out of nine job characteristics for Black individuals. Obtaining a Bachelor’s degree was associated with a decrease in the importance of sensory abilities (−0.094, *p* < 0.01), work output (−0.197, *p* < 0.001), and physical work conditions (−0.433, *p* < 0.001) for Black individuals. In some cases, having a Masters or PhD increased the parameter estimate, i.e., for work output, interacting with others, resource management skills, and interpersonal relationships. However, for Whites, these associations were not significant.

## 4. Discussion

The primary goal of this study was to advance understanding of work as a social determinant of racial inequalities in health. Systematic research focused on work and employment as a social determinant of racial health disparities is underdeveloped, as evidenced by the U.S. National Institutes of Health’s issuance of a funding opportunity announcement in the fall of 2021. To achieve the primary goal, we leveraged data from the Occupational Information Network (O*NET), a comprehensive source of objective information about multiple features of jobs, and a national sample enriched to study racial inequalities in health [39]. Although our study was conceptually informed by Warr’s Vitamin Model [20] and common mechanisms of inequality, the results of this study should be interpreted as exploratory because the O*NET data are not designed to study human health. Despite the study’s exploratory nature, the results of this analysis make several contributions to the literature.

The results of this study indicated that three of nine studied job characteristics systematically differ by race. Specifically, we found higher technical skills, resource management skills, and structural job characteristics in jobs held by non-Black individuals relative to Black individuals. These results are consistent with a previous qualitative review of the literature reporting that Black individuals and other minorities have less access to control in their work relative to White individuals and greater exposure to psychological demands [29]. The current results extend previous results, based on predominantly self-reported job characteristics, by demonstrating a similar pattern using external characterizations of jobs captured by the O*NET. Although modest in number, the results provide robust evidence that Black individuals have less opportunity to acquire and hone advanced skills through their work, including higher-level decision-making, linked with a wide variety of health outcomes, such as hypertension and cognitive functioning [40]. Importantly, we found this evidence of differential exposure controlling for common human capital indicators like educational attainment, age, and sex.

We found evidence of differential gain by race in the putative effects of job characteristics on eudemonic well-being. Eight of the nine job attributes studied were associated with at least one dimension of well-being, and 84% of those associated differed by race. Consistently, job characteristics had either benign or negative associations with well-being among Black individuals but positive associations with well-being among non-Black individuals. The consistency of the observed pattern of results makes two critical contributions to the literature. First and most simply, the results suggest that Black individuals’ psychological functioning and their subsequent resilience [35] is less work-centric than non-Black individuals’ (primarily White individuals’), suggesting that job characteristics are more beneficial to non-Black individuals compared to Black individuals. Interestingly, this result is consistent with Cundiff and Matthews [41] meta-analytic results indicating a substantially weaker association of perceived socioeconomic status with health among Black individuals than White individuals. Perhaps job attributes such as those captured by the O*NET play a more meaningful role in White individuals’ conceptions of both social standing than Black individuals’. Second, and more importantly, the focus on eudemonic well-being raises the notable question of whether current conceptions of work, such as NIOSH’s “organization of work” model [18] and other common models, including the Vitamin Model [20] or the Job Demands-Control Model [19], are based on ethnocentric concepts that give hegemonic advantage to White individuals. The models were ostensibly created to help create “healthy work” arrangements for all; therefore, the most immediate test of model accuracy would be workers’ ability to function (i.e., eudemonic well-being). Subsequent research could then consider indicators of disorder (e.g., depression or anxiety symptoms) and indicators of disease. The potential for implicit biases favoring White-centric conceptions of “healthy work” demands ongoing attention as the study of work as a social determinant of health continues to unfold.

This study found little evidence of curvilinear health effects of job characteristics. Specifically, of over 108 tested curvilinear associations, only 6 (5.5%) were statistically significant—no more than would be expected by chance alone. Not discounting the possibility that existing models of work and health may have implicit biases that favor White individuals, the relative lack of evidence supportive of Warr’s Vitamin Model [18] should not be overinterpreted due to a weakness of the O*NET data. That is, values of job characteristics are identical for everyone with the same Standard Occupational Classification, regardless of potential variability in experience across industry sectors or employers. For example, all “Registered Nurses” were given the same value for every job characteristic considered in this study. However, the daily work experiences of Registered Nurses working in a school infirmary are likely very different from Registered Nurses working in a trauma center or pediatric ambulatory clinic. Therefore, the restricted range of scores likely attenuated the ability to detect possible nonlinear effects.

Finally, the differential gain of education for well-being by race was surprising and requires comment. We found advanced educational attainment benefited multiple dimensions of well-being among Black individuals but had benign or negative implications for non-Black individuals, after controlling for age, gender, race, and job characteristics. Perhaps this observed difference is an example of moderated mediation, which we did not test. Perhaps advanced education benefits well-being through (i.e., mediated by) job characteristics among non-Black individuals resulting in attenuated direct associations of education with well-being. By contrast, among Black individuals, perhaps job characteristics have a weaker ability to mediate the education and well-being association, resulting in larger effects for education. Moderated mediation seems plausible given some evidence that graduate degrees benefit non-Black individuals but not Black individuals for some job characteristics (see Table 2) and is worthy of additional future research.

### Strengths and Limitations

This study utilized the newly merged and published O*NET data, which incorporates US-based occupational job characteristics on thousands of jobs. This study also advanced the investigation of work as a social determinant of health, by investigating the six dimensions of well-being. The six dimensions of well-being is a eudemonic measure of well-being, which advances the commonly investigated hedonic measures, which traditionally investigated mainly depression and anxiety. Furthermore, this study used data from a large nationally representative health and well-being dataset, including an oversample of Black individuals from Milwaukee, WI. A limitation to this study is that we only considered the *importance* variable of the job characteristic. O*NET also includes information on the *level* of the job characteristic required for job performance.

## 5. Conclusions

This study found support for Black individuals’ differential exposure and diminished gain with regards to job characteristics and psychological well-being, compared to White individuals, using data from a national study of U.S. adults merged with occupational data on over 900 U.S. jobs. The findings suggested that Black individuals were differentially exposed to fewer occupational opportunities, as evidenced by White individuals’ ability to obtain jobs wherein certain job characteristics were more important for overall job performance. Furthermore, the importance of job characteristics to job performance was found to be more beneficial for non-Black individuals’ psychological wellbeing compared to Black individuals. Race modified the pathway to health through education. Furthermore, we found support for Black individuals’ diminished gain with regards to education, suggesting that the benefit of education for obtaining jobs and achieving good health is systematically smaller for Black individuals compared to non-Black individuals. Overall, this study highlighted racial health inequalities with regards to Black individuals, who are systematically more challenged to obtain jobs with certain occupational characteristics, and have a harder time converting education into good jobs, as compared to non-Black individuals.

## Figures and Tables

**Table 1 ijerph-19-09820-t001:** Descriptive statistics.

	Non-Black Individuals (n = 1781)			Black Individuals (n = 396)			
Variable	N	%	Mean (SD)	N	%	Mean (SD)	*p*
Age (range 23–75)	1781		45.4 (12.2)	396		42.4 (11.2)	0.017
Sex	1781			396			<0.001
Male	928	52.1		162	40.9		
Female	853	47.9		234	59.1		
Education	1779			395			<0.001
HS/GED or less	308	17.4		125	31.6		
Some College	532	29.9		152	38.4		
Bachelor’s Degree	510	28.6		64	16.2		
Masters or PhD	429	24.1		54	13.6		
Autonomy	1259		36.3 (6.89)	229		38.4 (6.66)	<0.001
Environmental Mastery	1259		36.9 (7.37)	229		37.4 (7.18)	0.298
Personal Growth	1259		39.1 (6.4)	229		40 (6.5)	0.05
Positive Relations with Others	1259		39.2 (7.2)	229		39.8 (7.02)	0.268
Purpose in Life	1263		38.8 (6.69)	229		40.1 (6.53)	0.006
Self-acceptance	1259		37.1 (8.34)	229		37.9 (8.13)	0.165
Pre-tax income ($)	1589		61,648.04 (46,488.90)	344		39,473.40 (31,643.38)	<0.001

**Table 2 ijerph-19-09820-t002:** Results from Generalized Estimating Equations investigating racial and educational variation in job characteristics, where Black race and Less than High School Education are the reference groups.

	Sensory Abilities	Information Input	Work Output	Interacting with Others	Technical Skills	Resource Management Skills	Interpersonal Relationships	Physical Work Conditions	Structural Job Characteristics
Race (B)	0	0	0	0	0	0	0	0	0
Race (NB)	0.065	0.026	0.03	0.097	0.117 *	0.113 *	0.099	−0.046	0.069 *
Education (Less than High School)	0	0	0	0	0	0	0	0	0
Some College	0.026	0.077	0.017	0.172 ***	0.036	0.122 **	0.21 ***	−0.066	0.093 **
Bachelor’s Degree	−0.094 **	0.002	−0.197 ***	0.365 ***	−0.055	0.356 ***	0.373 ***	−0.433 ***	0.185 ***
Masters or PhD	−0.057	0.01	−0.235 ***	0.461 ***	−0.036	0.472 ***	0.505 ***	−0.393 ***	0.122 **
Age	−4.204 × 10^−5^	−0.001	0.0	0.0	−0.001	0.001	−8.8 × 10^−5^	0	0
Sex	−0.136 ***	−0.155 ***	−0.238 ***	0.083 ***	−0.333 ***	−0.105 ***	0.076 ***	−0.215 ***	−0.103 ***
Sample	0.004	−0.015	−0.06	−0.042	−0.032	−0.055	−0.027	−0.053	−0.042
Race(NB) * Some College	−0.067	−0.002	−0.04	−0.019	−0.027	0.049	−0.058	−0.048	−0.022
Race(NB) * Bachelor’s Degree	−0.056	−0.026	−0.06	−0.081	−0.092	−0.073	−0.08	0.057	−0.11 *
Race(NB) * Masters or PhD	−0.105 *	−0.014	−0.115	−0.047	−0.175 *	−0.135	−0.102	−0.086	−0.03

* *p* < 0.05; ** *p* < 0.01; *** *p* < 0.001; B = Black; NB = Non-Black.

**Table 3 ijerph-19-09820-t003:** Summary table indicating the linear and curvilinear associations for which race was a statistically signficant moderator of job characteristics and Ryff’s Six Dimensions of Well-Being.

	Autonomy		Environmental Mastery		Personal Growth		Positive Relations		Purpose		Self-Acceptance	
	Linear	Curvilinear	Linear	Curvilinear	Linear	Curvilinear	Linear	Curvilinear	Linear	Curvilinear	Linear	Curvilinear
Sensory Abilities	Yes	No	No	No	No	No	Yes	No	No	No	Yes	No
Information Input	No	No	No	No	No	No	No	No	No	No	Yes	No
Work Output	Yes	No	No	No	No	No	No	No	No	No	No	No
Interacting with Others	No	No	No	No	No	No	No	No	No	Yes	No	Yes
Technical Skills	No	No	No	No	No	No	No	No	No	No	No	No
Resource Management Skills	No	No	No	No	No	No	No	No	Yes	No	Yes	No
Interpersonal Relations	No	No	No	No	No	No	No	No	No	No	No	No
Physical Work Conditions	Yes	Yes	Yes	No	Yes	No	No	No	No	No	Yes	No
Structural Job Characteristics	No	No	No	No	No	No	No	No	No	No	No	No

**Table 4 ijerph-19-09820-t004:** Summary table indicating for which associations race (Non-Black) was a statistically signficant moderator of education on Ryff’s Six Dimensions of Well-Being.

	Autonomy	Environmental Mastery	Personal Growth	Positive Relations	Purpose	Self-Acceptance
**Sensory Abilities**						
Race(Non-Black) * Some College	No	No	No	No	No	No
Race(Non-Black) * Bachelor’s Degree	Yes	Yes	No	No	Yes	Yes
Race(Non-Black) * Masters or PhD	Yes	No	No	No	No	No
**Information Input**						
Race(Non-Black) * Some College	No	No	No	No	No	No
Race(Non-Black) * Bachelor’s Degree	Yes	Yes	Yes	No	Yes	Yes
Race(Non-Black) * Masters or PhD	Yes	No	No	Yes	No	No
**Work Output**						Yes
Race(Non-Black) * Some College	No	No	No	No	No	No
Race(Non-Black) * Bachelor’s Degree	Yes	No	Yes	No	Yes	Yes
Race(Non-Black) * Masters or PhD	Yes	No	No	No	No	No
**Interacting with Others**						
Race(Non-Black) * Some College	No	No	No	No	Yes	No
Race(Non-Black) * Bachelor’s Degree	Yes	Yes	Yes	No	Yes	Yes
Race(Non-Black) * Masters or PhD	Yes	No	No	Yes	Yes	Yes
**Technical Skills**						
Race(Non-Black) * Some College	No	No	No	No	No	No
Race(Non-Black) * Bachelor’s Degree	Yes	Yes	Yes	No	Yes	Yes
Race(Non-Black) * Masters or PhD	Yes	No	No	No	No	No
**Resource Management Skills**						
Race(Non-Black) * Some College	No	No	No	No	Yes	No
Race(Non-Black) * Bachelor’s Degree	Yes	Yes	Yes	No	Yes	Yes
Race(Non-Black) * Masters or PhD	Yes	No	No	No	Yes	Yes
**Interpersonal Relations**						
Race(Non-Black) * Some College	No	No	No	No	Yes	No
Race(Non-Black) * Bachelor’s Degree	Yes	Yes	No	No	Yes	Yes
Race(Non-Black) * Masters or PhD	Yes	No	No	No	Yes	No
**Physical Work Conditions**						
Race(Non-Black) * Some College	No	No	No	No	No	No
Race(Non-Black) * Bachelor’s Degree	No	No	No	No	Yes	No
Race(Non-Black) * Masters or PhD	Yes	No	No	No	No	No
**Structural Job Characteristics**						
Race(Non-Black) * Some College	No	No	No	No	No	No
Race(Non-Black) * Bachelor’s Degree	Yes	Yes	Yes	No	Yes	Yes
Race(Non-Black) * Masters or PhD	Yes	No	No	No	No	No

* Interaction between variables.

## Data Availability

All data used in this study are publicly available. MIDUS Refresher, Milwaukee Refresher and corresponding O*NET datasets are available through the MIDUS colectica https://midus.colectica.org/ (accessed on 28 March 2022).

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
