# Peer review of "Work as a Social Determinant of Racial Health Inequalities"

_ijerph, 2022, doi:10.3390/ijerph19169820_

Round 1
Reviewer 1 Report
The description of the sample is thin: the authors don’t indicate the overall “n” or the percentage of the sample that was African-American. This section of the article needs to be expanded to give the reader a clearer understanding of the nature of the sample.
The description of the sample indicates a HUGE gap in income between Black and non-Black respondents (the former earned almost as much, on average, as the Black sample). It does not appear that this was controlled for in the analysis (or at least there’s no discussion of the effect of income on outcomes). It is possible that controlling for income might have an effect on the observed results. If possible, this should be incorporated into the analysis.
The analysis focuses only on mental health outcomes. It’s not clear why the authors chose this focus – the introduction does not have much to say about this, apart from two lines on p. 3 (lines 116 and 117) with a reference to a study conducted 20 years ago. This seems a thin basis for narrowing the dependent variable to this extent, especially since the introduction spends considerable time talking about other aspects of health and their potential relationship to worker abilities and skills. More needs to be said about the choice to focus on mental health outcomes, rather than other aspects of health.
Finally, there is considerable discussion of Warr’s “vitamin model,” which appears to derive from a paper published in 1994. Since the paper relies heavily on this article for its conceptual framework (especially the emphasis on potentially curvi-linear relationships), a more thorough summary of that article would make the analysis clearer.
Author Response
Thank you for your helpful comments and suggestions for the manuscript. A point by point response to each of your queries has been uploaded below.

Reviewer 2 Report
I was pleased to read the manuscript entitled "Work as a Social Determinant of Racial Health Inequalities" and to review it.
This study aimed to advance understanding about work as a social determinant of racial inequalities in health. Judging from the objectives of the research and the results obtained in the study, the article is suitable for section Occupational Safety and Health, Work Organization, Occupational Stress, and Mental Health and Wellbeing: Advances in the Evidence and Approaches to Intervention.
The article is written in a typical format.
The rationale of the study is well described.
Materials and Methods:
1. It is necessary to write briefly how the sample was formed, how many respondents were selected, what was the response rate.
2. Lines 153-160: Explain the abbreviations MR and MKER (In total, it is desirable to provide a list of all abbreviations).
Nevertheless, while reading the Results section several questions arose and inaccuracies were noticed, which I recommend to fix not only personally for the reviewer but probably to readers too:
1. Table 2: It is unclear which groups are compared or which group is the control.
2. Tables 3 and 4: There is no clear link between the data in the tables and their description in the text (e.g." The highlighted boxed indicate the associations for which race was a statistically significant moderator" - what does it mean?). Readers who are not familiar enough with the analysis of Generalized Estimating Equations should be aware of what is presented in the tables. The titles of the tables need to be clarified and explanations of the data need to be provided in text and footnotes.
References: Provide descriptions of references according to the requirements of the journal.
Thank you for considering my opinion. I encourage authors to keep on working to improve the manuscript.
Author Response

(The authors gave the same response as above.)

Round 2
Reviewer 1 Report
The authors have said more about the sample size and composition and added material on their decision to focus on mental health (not physical health).
I don't fully agree that the authors have addressed the effect of income in their model. While education correlates strongly with income, it's not the same thing. The economic returns to education are widely known to be lower for Black than non-Black Americans, so looking at the effect of income separately from the effect of education might shed light on the racial differences the authors are exploring. The authors summarize one of their findings as follows: "Job characteristics had either benign or negative associations with well-being among black individuals but consistently positive associations with well-being among non-black individuals." It is at least possible that lower income among black respondents contributed to the differences found.
I acknowledge that the authors don't agree. Perhaps a solution would be to state, explicitly, in the text of the analysis how income is or is not being considered in the analysis (and why).
